# Comparison of static and dynamic cerebral autoregulation under anesthesia influence in a controlled animal model

**Alexander Ruesch**[1], **Deepshikha Acharya**[1], **Samantha Schmitt**[1,2,3], **Jason Yang**[1], **Matthew A. Smith**[1,2,3], **Jana M. Kainerstorfer**[1,2]*

**1** Department of Biomedical Engineering, Carnegie Mellon University, Pittsburgh, Pennsylvania, United States of America, **2** Neuroscience Institute, Carnegie Mellon University, Pittsburgh, Pennsylvania, United States of America, **3** Department of Ophthalmology, University of Pittsburgh, Pittsburgh, Pennsylvania, United States of America

* jkainers@andrew.cmu.edu

**Data Availability Statement:** All raw data files used to create results in this publication are provided on Figshare (DOI: 10.6084/m9.figshare.12937505).

## Abstract

The brain's ability to maintain cerebral blood flow approximately constant despite cerebral perfusion pressure changes is known as cerebral autoregulation (CA) and is governed by vasoconstriction and vasodilation. Cerebral perfusion pressure is defined as the pressure gradient between arterial blood pressure and intracranial pressure. Measuring CA is a challenging task and has created a variety of evaluation methods, which are often categorized as static and dynamic CA assessments. Because CA is quantified as the performance of a regulatory system and no physical ground truth can be measured, conflicting results are reported. The conflict further arises from a lack of healthy volunteer data with respect to cerebral perfusion pressure measurements and the variety of diseases in which CA ability is impaired, including stroke, traumatic brain injury and hydrocephalus. To overcome these differences, we present a healthy non-human primate model in which we can control the ability to autoregulate blood flow through the type of anesthesia (isoflurane vs fentanyl). We show how three different assessment methods can be used to measure CA impairment, and how static and dynamic autoregulation compare under challenges in intracranial pressure and blood pressure. We reconstructed Lassen's curve for two groups of anesthesia, where only the fentanyl anesthetized group yielded the canonical shape. Cerebral perfusion pressure allowed for the best distinction between the fentanyl and isoflurane anesthetized groups. The autoregulatory response time to induced oscillations in intracranial pressure and blood pressure, measured as the phase lag between intracranial pressure and blood pressure, was able to determine autoregulatory impairment in agreement with static autoregulation. Static and dynamic CA both show impairment in high dose isoflurane anesthesia, while low isoflurane in combination with fentanyl anesthesia maintains CA, offering a repeatable animal model for CA studies.

**Funding:** The authors acknowledge the financial support for the research, authorship, and publication through the Center for Machine Learning and Health (CMLH) to AR (https://www.cs.cmu.edu/cmlh-cfp), Pennsylvania Infrastructure Technology Alliance (PITA) to JK (https://dced.pa.gov/programs/pennsylvania-infrastructure-technology-alliance-pita/), the American Heart Association (AHA) 17SDG33700047 to JK (https://pureresearch.americanheart.org/en/), and the National Institutes of Health (NIH) R21-EB024675 to JK (https://www.nih.gov/). The funders had no role in study design, data collection and analysis, decision to publish, or preparation of the manuscript.

**Competing interests:** The authors have declared that no competing interests exist.

## Introduction

The human brain resides inside an enclosed environment formed by the meninges and skull, encasing it in a unique pressure environment in which blood perfusion is not only determined by arterial blood pressure (ABP), but also by intracranial pressure (ICP). More specifically, ICP modulates the venous sinus pressure that determines the blood return to the heart. The pressure gradient from arterial to venous pressure is the driving force for blood flow. Similarly in the skull, venous sinus pressure is approximately equal to ICP, which leads to the clinically used simplification that cerebral perfusion pressure (CPP), which drives blood flow in the brain, can be estimated as the pressure difference between mean arterial blood pressure (MAP) and mean ICP [1]. To maintain stable blood flow despite changes in either ABP, induced for example by exercise, or ICP, induced by posture changes, the arteries and arterioles of the brain can dilate and constrict [1]. The vessel diameter change then allows to regulate cerebral blood flow (CBF), keeping it stable despite CPP changes. This mechanism is known as cerebral autoregulation (CA) and its dependence on pressure has been described as early as the 1950s by Lassen et al. [2].

The ability to regulate blood flow locally in the brain can be impaired in many diseases, including traumatic brain injury [3, 4], hydrocephalus [5], and stroke [6]. In some cases, the perfusion of the brain is regulated by clinicians through pressure manipulation, thus CPP is maintained in a literature-based range to improve CBF. Current treatment guidelines for traumatic brain injury suggest to maintain CPP above 60–70 mmHg in adults [7] and above 40–50 mmHg in children [8]. An alternative method, that has shown potential to improve patient outcome in traumatic brain injury, is to measure the CA impairment through evaluation of pressure reactivity and to adjust CPP accordingly [3, 9, 10].

The clinical need to measure CA is well established but disagreement about the consistency between measurement methods exists [11–13]. CA measurement methods are commonly separated into two groups: static and dynamic CA [14]. Static CA, sometimes referred to as steady-state CA, are autoregulatory responses to steady-state changes in ABP or ICP, giving information about the range of CPP in which CA is active [1]. Dynamic CA refers to rapid changes, typically in ABP, where the response time in blood flow recovery is indicative of CA [15]. While the two groups are related, differences [12] and similarities [11] between them have been reported. Static autoregulation is described by Lassen's curve [2, 16], which is constructed by plotting steady state (baseline) CBF vs. CPP. This curve shows a characteristic shape of monotonically increasing CBF with CPP, and a distinct CBF plateau in a range of CPP where autoregulation is intact and vessel constriction can compensate for increased pressure. Alternatively, under the assumption that ICP remains constant or changes are negligible, Lassen's curve has been reported based on mean ABP changes vs. CBF [2]. Dynamic autoregulation is described based on transient changes in CBF in response to pressure changes. One can induce a step function change in ABP through thigh cuff occlusion, which has been done before to describe the autoregulatory response time [17]. Thigh cuff inflations can however cause discomfort to patients. Another example for dynamic autoregulation assessment is to induce oscillations for example into ABP via paced breathing or modulation of the positive end-expiratory pressure (PEEP) in the lungs. PEEP describes the positive pressure that is maintained inside the trachea and lungs after exhalation in positive pressure respiration ventilators. Fraser et al. [18] and Kainerstorfer et al. [19] showed that a cutoff frequency ($f_c = 0.03$ Hz) can be found above which the CA response decreases. Below $f_c$, CA can be measured via the phase difference of ABP and ICP [18] or modeled by a high-pass filter scaled with the Grubb's exponent [20] that relates blood flow and blood volume changes [19]. Pressure reactivity measurements are another method to describe autoregulation intactness and was

introduced by Czosnyka et al. [21], where a moving Pearson correlation between ICP and ABP during naturally occurring changes of pressures within a 5 minute window describes pressure reactivity, known as the pressure reactivity index (PRx). If PRx is low (<0.3) or negative, the autoregulatory system is intact, while high correlations show impairment. PRx assumes that a change in ABP will lead to vasoconstriction or vasodilation and that this effect is significant enough to change ICP as the blood volume in the brain is changed. An increase in ABP, for example, leads to vasoconstriction. The smaller volume of blood resulting from the constriction then reduces ICP, making ABP and ICP negatively correlated. If CA is impaired and vascular diameter change is no longer possible, the system becomes passive and ICP will rise as ABP rises, hence the PRx value will be positive. This method has been used to determine the optimal CPP ($CPP_{opt}$), which is where PRx is the smallest, in order to optimize CA intactness and patient outcome [22]. The calculation of $CPP_{opt}$ shows a range of CA intactness, which is a static CA characteristic. At the same time, the dynamic PRx assessment allows to determine a degree of CA impairment. Therefore, PRx is a hybrid method and here referred to as *a* pseudo-dynamic CA.

Derivatives of the above described static, dynamic, and pseudo-dynamic measurements have been proposed in order to reduce the need of invasive ICP sensors and improve reliability. Such methods often replace ICP or ABP with other hemodynamic measurements such as hemoglobin concentrations, tissue oxygenation or blood volume and summaries of them can be found elsewhere [15, 23–25], including non-invasive diffuse optical methods used in this article [26, 27]. Despite many decades of research and a large variety of measurement methods, a consensus on the effectiveness and use of CA measurements to guide clinical treatment has not yet been reached. We believe that common challenges in CA assessment are the comparison to Lassen's curve as a point of reference and the unavailability of ICP measurements in healthy volunteers due to the highly invasive acquisition. These challenges necessitate clinical studies to rely on data from patients with severe injuries and diseases that might affect the autoregulatory system in unpredictable ways and make comparisons across institutions with varying guidelines for treatment difficult.

We want to overcome these challenges of current CA studies and to create a basis for CA assessments by showing the effects of ABP and ICP perturbations on healthy non-human primates (NHP) with controlled CA impairment through isoflurane anesthesia. Inhaled and intravenous anesthetic effects on cerebral physiology can differ strongly, which can be translated to their effect on vasodilation and cardiac output modulations in a dose related manner, as well as cerebral metabolic rates [28]. Isoflurane, as an example for volatile anesthetics, has a cerebrovascular dilatory effect, which leads to an increase in CBF and cerebral blood volume (CBV) [28], while at the same time reducing ABP significantly, and thus cardiac output [29]. This forced dilation in cerebral vasculature is presumably the reason why CA is no longer able to control blood flow. Intravenous anesthetics, especially opioids such as Fentanyl, have generally smaller effects on cardiac output and do not cause vasodilation or vasoconstriction [29], which will largely maintain CA ability. Isoflurane and fentanyl anesthetics are therefore good candidates to compare effects of CA and the effects of regulatory vasodilation on CBF maintenance. With this work investigate how anesthetics can be used to manipulate CA in non-human primates, and how this generates a healthy subject basis for CA studies. We further analyze the agreement in the measurement methods of Lassen's curve [2], Fraser's phase delays [18], and Czosnyka's pressure reactivity index [21], under conditions of ABP and ICP oscillations, and the importance of CPP measurements as compared to ABP measurements alone.

## Materials and methods

We compared a variety of measurements of autoregulation in a NHP model during intact and impaired autoregulation, induced both by elevated ICP and isoflurane anesthesia.

### Diffuse correlation spectroscopy

In order to measure CBF we used a custom-built diffuse correlation spectroscopy (DCS) system. Details about the specifics and DCS operation can be found elsewhere [30, 31] and are briefly summarized here. Optical fibers were placed on the exposed skull, with one fiber delivering long-coherence length laser light (785 nm or 850 nm wavelength). The returning light was captured by 4 bundled single-mode or few-mode detector fibers 2 cm away from the source light. Software correlation was used to measure temporal changes in speckle patterns in the perfused brain tissue. The autocorrelation of the light intensity was converted into electric-field autocorrelation. From here, a solution to the diffusion equation was then fitted to the curve, yielding the diffusion coefficient, $\alpha D_b$. This term describes the diffusion of red blood cells in the brain and has previously been linked to and correlated with cerebral blood flow [32–34]. The $\alpha$ term describes the ratio of moving scattering particles (red blood cells) to static particles (such as blood vessels, brain and muscle tissue at rest). $D_b$ describes the Brownian motion diffusion coefficient, which is a direct derivative of the Brownian motion model used to derive the solution to the diffusion equation for scattering light in tissue. Because $\alpha$ is largely constant, its influence on the rate of blood flow change is negligible, such that $\alpha D_b$ can be treated as one variable in the fitting algorithm. We refer to percentage changes of $\alpha D_b$ as $\Delta$CBF from here on, where $\Delta CBF = (\alpha D_b - \alpha D_{b,0})/\alpha D_{b,0}$, and $\alpha D_{b,0}$ is the flow at the beginning of the measurement.

### Experimental setup

All procedures were approved by the Institutional Animal Care and Use Committee of the University of Pittsburgh (#18093463) and complied with guidelines set forth in the National Institute of Health's Guide for the Care and Use of Laboratory Animals (2011). The facilities at the University of Pittsburgh are accredited by the Association for Assessment and Accreditation of Laboratory Animal Care International (AAALAC) and in compliance with the Standards for Humane Care and Use of Laboratory Animals of the Office of Laboratory Animal Welfare (OLAW D16-00118), including enrichments like treats, foraging, toys, visual and auditory stimulation, and human interaction. Furthermore, this manuscript is in compliance with the Animal Research: Reporting In Vivo Experiments (ARRIVE) guidelines. Measurements were taken on a group of 12 NHPs (Macaca mulatta, f/m: 0/12, 8.1 ± 1.7 years, 9.9 ± 2.5 kg). All animals have been part of previous scientific experiments, before they were used in this study, following the guidelines to reduce the number of animals in scientific experiments to a minimum and maximize their use. Due to these limitations, we were not able to influence the age and sex distribution of the subject pool and can therefore not assess these variables in our study. The NHPs were initially sedated with 20 mg/kg of Ketamine independently of, or in combination with 1 mg/kg Diazepam, and 0.04 mg/kg of Atropine in their home cage and transported to the operating room. There, NHPs were intubated and maintained under anesthesia using either 1–3% of isoflurane ($N_{Isoflurane}$ = 7) or a combination of intravenously administered fentanyl at 10–25 μg/kg/hr and a minimal amount of isoflurane gas (< 1%) ($N_{Fentanyl}$ = 5). Anesthesia levels were adjusted throughout the experiment to ensure sufficient anesthesia depth as indicated by either a rise in ABP for isoflurane anesthesia, or a spike in heart rate after a toe-pinch test in fentanyl anesthetized NHPs. Isoflurane is known to suppress ABP and changes in ABP [29], which also is indicative of CA suppression. To monitor

sufficient anesthesia, heart rate, respiration rate, pulsatile arterial blood oxygenation (SpO2), end tidal carbon dioxide partial pressure (ETCO2), and other vital signs were recorded at 2 samples per minute through the SurgiVet veterinary monitor (Smiths Medical, Minneapolis, MN, USA). During our experiments, the animals were paralyzed with 0.1 mg/kg/hr of Vecuronium Bromide paralytic. The paralytic was necessary for a parallel experiment we ran in the same subjects. The infusion of paralytic at a constant dose was not expected to interfere with the autoregulatory assessment. NHPs were positioned with their heads facing forward in a stereotaxic apparatus, resting with their abdomen on the table. Body temperature was regulated by a water circulating heating pad.

An arterial line was placed in the carotid artery and ABP was recorded at a sampling rate of 100 Hz by an MPR1 Datalogger (Raumedic Helmbrechts, Germany). To measure and manipulate ICP, two small craniotomies were performed by drilling holes into the skull. A catheter (Lumbar catheter, Medtronic, Minneapolis MN) was placed into the lateral ventricle in the brain and connected to a saline reservoir. The height of the reservoir relative to the head was used to influence the pressure in the catheter, leading to fluid flow into the ventricle and therefore an ICP change. The second hole was used to place a parenchymal pressure sensor, also connected to the MPR1 Datalogger, recording ICP at a sampling rate of 100 Hz. The optical probe for ΔCBF acquisition was placed laterally and posterior to the craniotomies, near anterior-posterior zero in stereotaxic coordinates (Fig 1A).

The acute studies lasted 16.2 hours on average (SD 5.1 h, range 10.25 to 24.5 h). All of these studies were terminal because of the irreversible damage caused by extreme ranges of intracranial pressure. At the end of the acute study, animals were euthanized either by injection (N = 4) of a commercial euthanasia solution (Beuthanasia (0.22 ml/kg) or exsanguination via suction applied to the arterial line (N = 8) after deep anesthesia was insured with an elevated level of isoflurane (preceded by an injection of 300 units/kg heparin). The euthanasia procedures and study design were reviewed and approved by the Institutional Animal Care and Use Committee of the University of Pittsburgh (approval number 18093463) and complied with the American Veterinary Medical Association (AVMA) Guidelines for the Euthanasia of Animals.

## Experimental design

Steady-state measurements of 30 minutes or more were recorded before any manipulation of ICP or ABP. Oscillatory perturbations in ICP were induced by rotating the saline reservoir attached to a lever on a motor at 5 frequencies with a magnitude of 5 mmHg. Slow frequencies of 0.025 Hz, 0.059 Hz, 0.017 Hz, 0.033 Hz, and 0.009 Hz were induced, which were aimed to be above and below the autoregulatory cut-off frequency [18, 19]. The frequency order was once randomized and then kept constant to distinguish effects of time under severe conditions from the induced frequency. Four oscillations were performed per frequency with exception of the fastest 0.059 Hz frequency which was induced for 8 oscillations. This allowed an adequate time period to enable frequency analysis of the signals. After each set of ICP oscillations, the ICP baseline was changed by raising or lowering the motor in height. The height difference of the reservoir relative to the head determined the new ICP level (Fig 1B). ICP was held at the initial baseline at first (approx. 3–6 mmHg) and then gradually increased to up to 40 mmHg. The ICP baseline level was aimed to be 3, 6, 9, 12, 15, 20, 30, and 40 mmHg. The step size and maximum ICP varied between NHPs based on their ability to compensate for the excessive fluid.

To induce oscillations in ABP, PEEP was oscillated using a programmable ventilator (EMV +, 731 Series, ZOLL Medical Corporation, Chelmsford, MA, USA) in subset of non-human

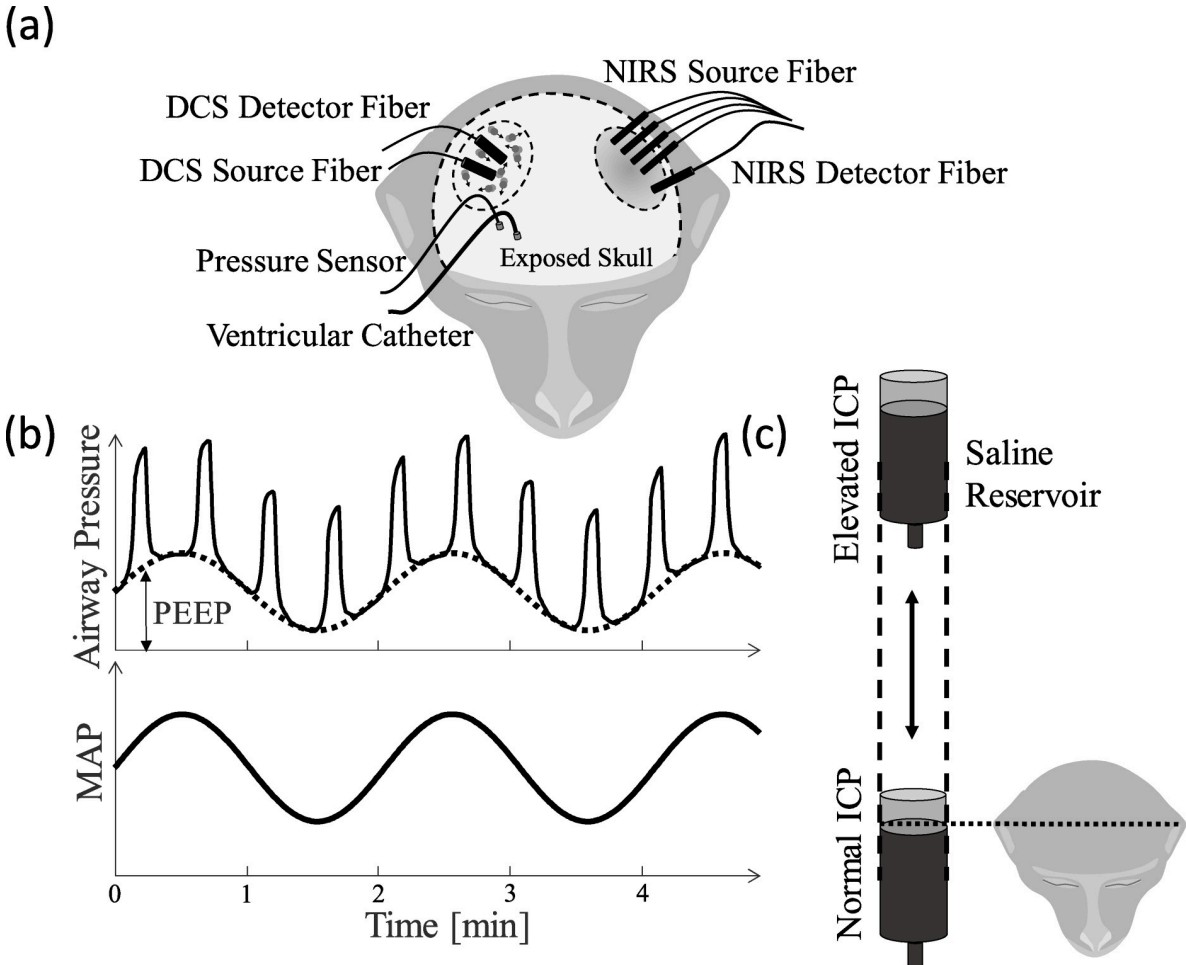

**Fig 1. Experimental setup.** (a) Location of DCS, catheter and ICP sensor relative to the NHP's head. These positions were maintained across all NHPs. (b) The saline reservoir connected to the ventricle catheter was raised and lowered in a circular motion on a lever, whose center point was placed relative to the NHP's head to allow for ICP manipulation. (c) Change in positive end-expiratory pressure (PEEP), the pressure remaining in the lungs after exhalation, through active programming of the ventilator influences systemic mean ABP (MAP).

primates ($N_{Fentanyl}$ = 5, $N_{Isoflurane}$ = 1) as shown in Fig 1C. The PEEP baseline value was set to 6 cmH$_2$O (8.1 mmHg) and oscillated at the same frequencies as ICP with a magnitude of 4 cmH$_2$O (5.4 mmHg). ABP oscillations were performed on every ICP baseline after the ICP oscillations in the same order of frequencies.

## Signal processing

To examine the effects of anesthesia and CPP on CA, we assessed static CA using Lassen's curve, dynamic CA using phase differences between ICP and ABP, and pseudo-dynamic CA using PRx. For all CA calculations the signals of ABP, ICP, and CBF were first aligned based on markers set during the experiment. MAP was calculated using the clinical standard of 2/3 diastolic ABP + 1/3 systolic ABP. CPP was then calculated as MAP–ICP. All signals were down sampled to 5 Hz.

**Static autoregulation.** In order to calculate Lassen's curve, all data were time averaged in bins of 10 seconds to remove effects of respiration and cardiac pulsation. The CBF data was cleaned from laser instabilities and artifacts by z-score rejection (z > 0.5) on an individual

NHP basis. Laser instabilities were identified based on the β value, defined as the y-axis intersection of the measured speckle pattern intensity auto-correlation in DCS, subtracted by 1. The fluctuations of the β value are expected to be very small and any larger deviation can be discarded as non-physiological noise. When laser stability was lost, presumably due to internal reflections of light at the laser to fiber interface, β started fluctuating significantly and that time point was identified as unstable by its z-score. A measurement was said to have artifacts if at any time point the β value was smaller than the median (approx. 0.1 for larger fiber core diameters and 0.5 for single mode fibers) by 0.01. CBF data were cleaned further by rejecting any averaged 10 second window of CBF outside a z-score range of 2 standard deviations, rejecting further artifacts from room light, motion, and laser instability not removed previously. CBF data was then time averaged within CPP bins of 1 mmHg width. A Lassen's curve was calculated for each NHP, based on the $\alpha D_b$ value. Individual Lassen's curves were then mean subtracted and mean divided for each NHP, to create ΔCBF, before they were averaged in groups of isoflurane and fentanyl anesthesia.

**Dynamic autoregulation.**   The phase difference between induced oscillations in ABP and ICP was calculated for assessing dynamic autoregulation in the frequency domain. To extract the phase information, narrow bandpass filters were generated using the Parks-McClellan finite impulse response filter generation algorithm in Matlab R2019a ("firpmord" and "firpm", The MathWorks Inc., Natick, MA, USA). This filter generates a pass band of 0.03 Hz width on either side of the induced frequencies such that the extracted signal is almost a perfect single frequency sinusoid. The Hilbert transform was applied, and magnitude and phase of the induced oscillations were extracted. Phase information was calculated independently for both ICP and ABP. Subtracting ICP phase from ABP phase yielded phase delay, which has previously been reported by Fraser et al. to be indicative of autoregulation [18]. We performed this calculation for oscillations based on fluid induction as well as PEEP. Phase delays per frequency were averaged for each anesthesia group.

**Pseudo-dynamic autoregulation.**   The PRx value was calculated based on Pearson's correlation coefficient of moving averaged ICP and ABP time traces. Signals were down sampled to 0.1 Hz by averaging 10 second periods of data. The Pearson's correlation between down sampled ICP and ABP was calculated over 5-minute-long periods. The same windows were used to calculate an average CPP. PRx values were sorted in 150 groups of CPP values between 0 and 180 mmHg. Before averaging the PRx values, the Fisher's Z transformation was applied to account for cosine shaped Pearson's correlations. The normal distributed Z values were then averaged, and the inverse transformation applied. This algorithm was applied to the two groups of different anesthesia protocols separately.

## Results

From a total of 12 NHP, we recorded CBF, ABP and ICP during varying pressure conditions in ICP and ABP. We collected a total of 45.6 hours of data in 7 isoflurane-anesthetized NHPs and a total of 60.5 hours in 5 fentanyl-anesthetized (with < 1% isoflurane) NHPs. During these measurements we successfully elevated ICP by fluid induction into the ventricles and were able to observe the reaction in ABP and CBF. Representative time traces for an animal under fentanyl anesthesia are seen in Fig 2.

The $ETCO_2$ value calculated as an average of the entire duration of the experiment on individual animals was showing no values outside an expected range [35, 36], indicating appropriate anesthesia (Fig 3).

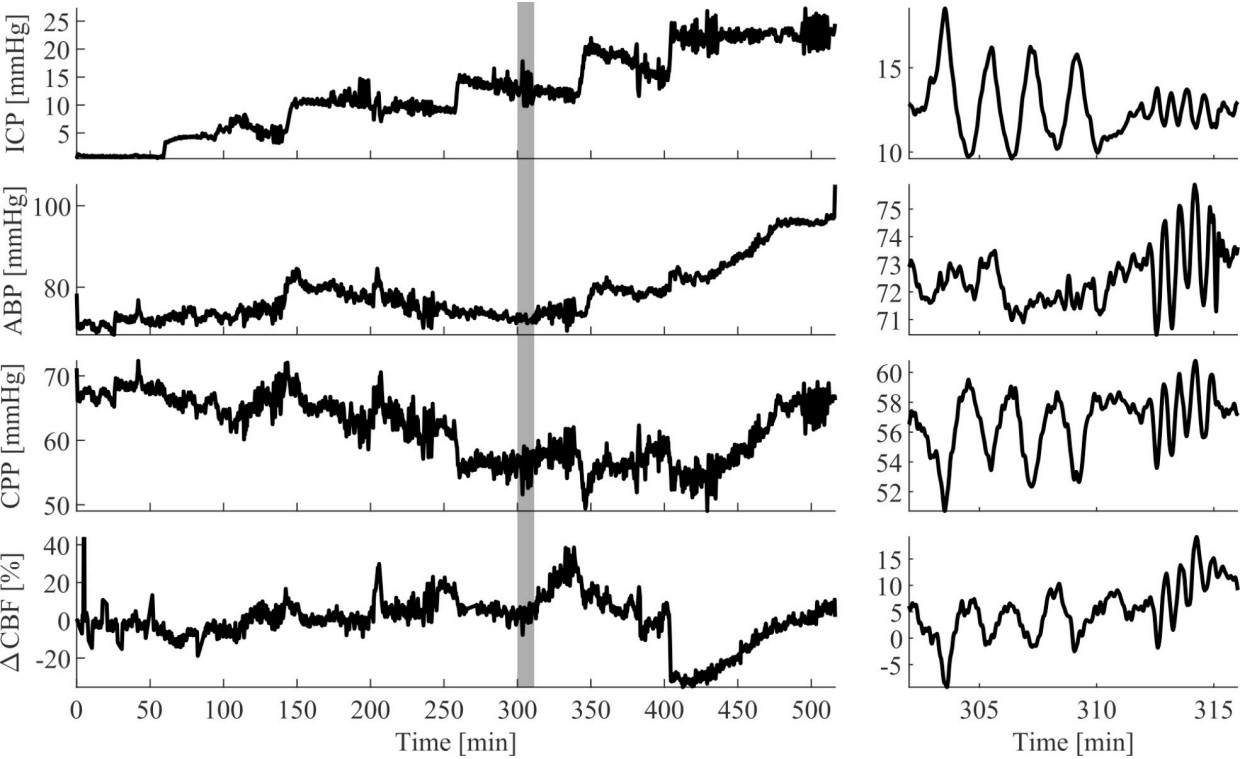

**Fig 2. A representative measurement of one NHP under fentanyl anesthesia.** The left side shows the entire time trace of this NHP. Clear steps in ICP are shown on the top graph. ABP follows the ICP baseline trend in the second graph from the top, keeping CPP in the third graph largely stable. CPP is defined as the difference between MAP and ICP. ΔCBF changes in the bottom graph show immediate reactions to baseline changes and recovery. The right-hand side shows a zoomed in frame around the minutes 300 to 317 (grey box on the left). Here it can be seen that ICP oscillations (until 310 minutes) as well as ABP oscillations (after 310 minutes) were translated into CPP and ΔCBF.

## Static autoregulation

Static autoregulation was assessed by Lassen's curve (Fig 4). Fentanyl-anesthetized NHPs show a plateau of ΔCBF between 60 mmHg and 100 mmHg, indicating intact CA. Sloped areas below and above the plateau, called lower (LLA) and upper (ULA) limit of autoregulation, showing CA impairment. On the other hand, isoflurane shows data dominantly below LLA, indicated by the slope and constantly low CPP in these NHPs. The histograms of CPP values show a wider spread in fentanyl than in isoflurane anesthesia, with overall higher values. The average CPP value and standard deviation for isoflurane were 48.4 ± 14.7 mmHg as compared to fentanyl with 85 ± 22.5 mmHg. We observed that CPP and ABP distributions were significantly different from each other while ICP values were similar. Given that the data were not normally distributed, we performed a Mann-Whitney U-test that evaluated if the two groups came from continuous distributions with different medians ('ranksum', Matlab R2019a, The MathWorks Inc., Natick, MA, USA). While the CPP distributions were significantly different from each other (p = 0.005), the histograms of ABP show a larger overlap region (p = 0.03). ICP for the same data are not significantly different (p = 0.4, Fig 4). Isoflurane levels were distinctly different between the isoflurane NHP group and the one with predominantly fentanyl as seen in Fig 4. The isoflurane levels reported here are the volume percentages of inhaled air supplied to the ventilator and were documented every 15 minutes. Thus, the histogram is shown as the number of measurements rather than data points, where every measurement corresponds to a baseline level of ICP and incorporates 45–90 minutes of data

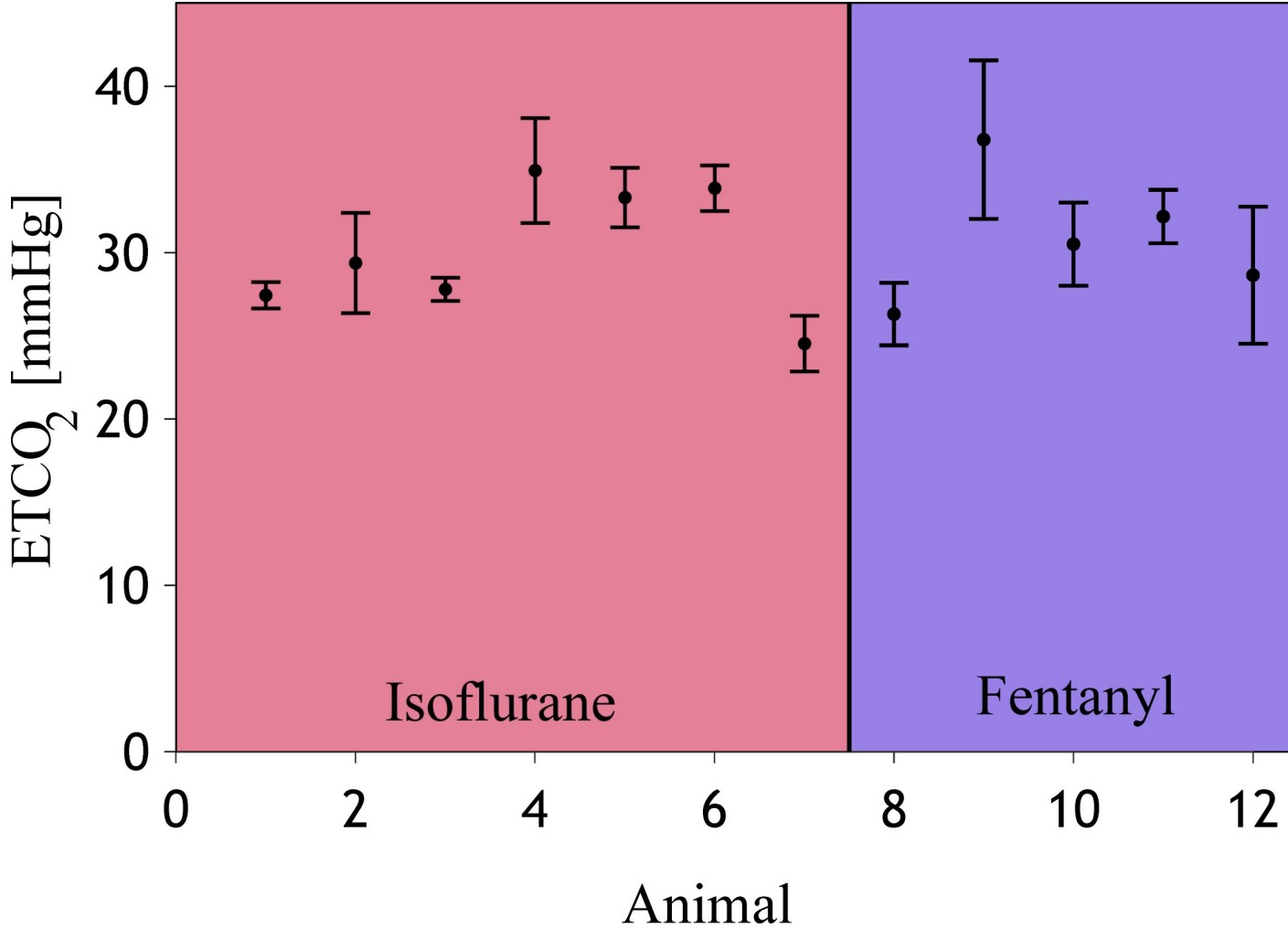

**Fig 3. Partial pressure in end tidal CO2 averages per animal.** The error bar shows the standard deviation around the mean ETCO2 calculated across all measurements of each individual animal. The left side highlighted in red shows isoflurane anesthetized animals, the right side shows predominantly fentanyl anesthetized subjects.

### Pseudo-dynamic autoregulation

The PRx values were calculated as the moving average between ABP and ICP on individual animal data. Group averages based on anesthesia regime are shown in Fig 5. The isoflurane-anesthetized NHPs (red) show an averaged PRx well above the zero-line at a broad range of CPPs, indicating CA impairment. We found that fentanyl-anesthetized NHPs (blue) had an averaged negative trend within the Lassen's curve plateau of 60 mmHg to 100 mmHg, indicating intact CA.

### Dynamic autoregulation

The phase delay between ABP and ICP was calculated during oscillation of ICP (based on fluid induction) and ABP (based on PEEP oscillations). ABP oscillations show that fentanyl anesthetized NHPs have a phase lag of approx. 180 degrees as seen in Fig 6, which has also been reported by Fraser et al. [18]. After the cutoff frequency around 0.033 Hz, the phase difference reduced to 90 degrees. Isoflurane maintained a phase difference below 90 degrees, indicating

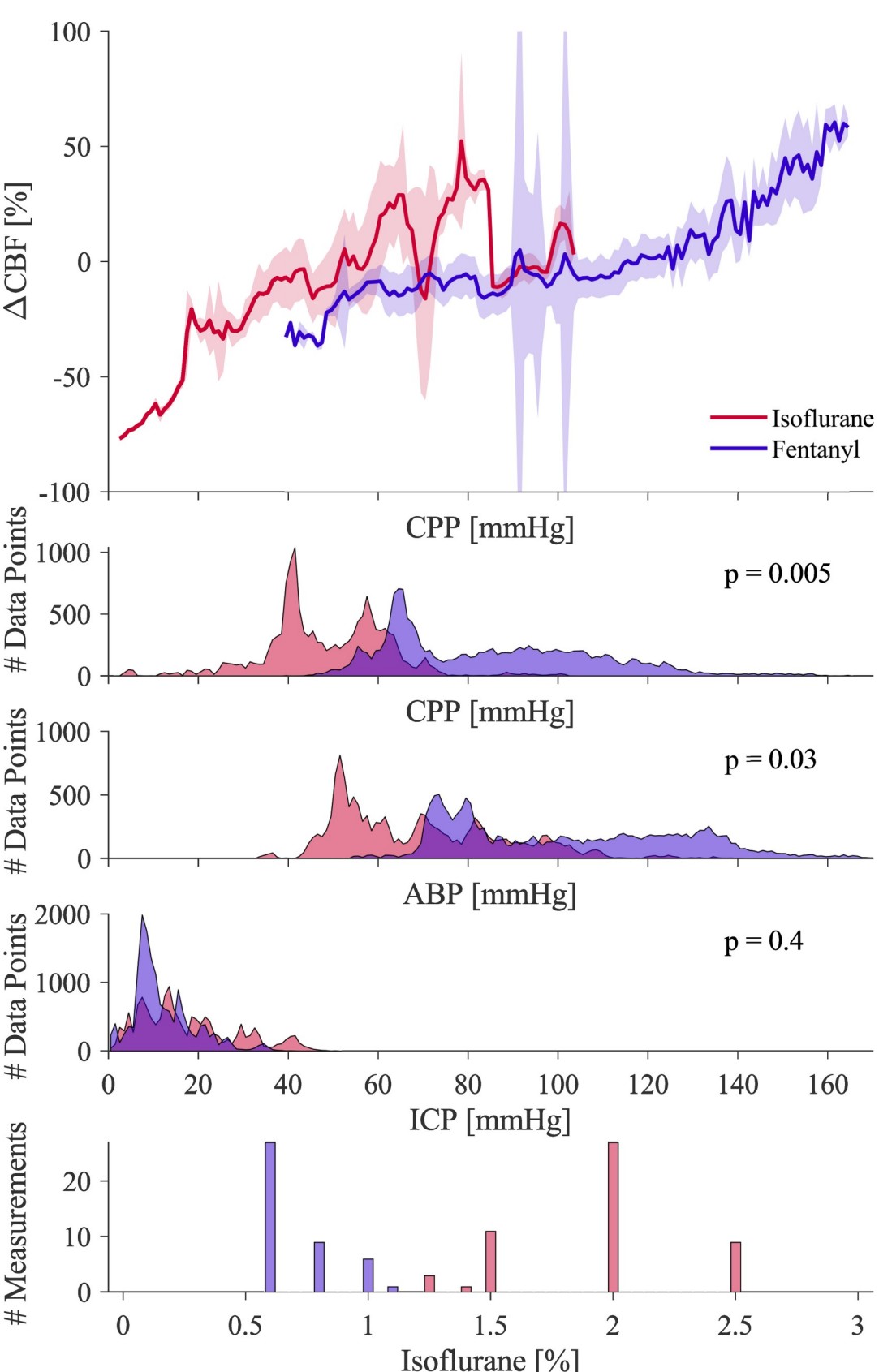

**Fig 4. Lassen's curve shows the relationship of ΔCBF over CPP (top) as a function of anesthetic.** Here the red line indicates isoflurane anesthesia, and the blue line represents the group with mainly fentanyl anesthesia. Shaded areas show the standard deviation across all NHPs in this group. The histograms are based on group averaged data for CPP, ABP, and ICP (from the top), with histogram bin sizes of 1 mmHg. A Mann-Whitney U-test was performed to test statistical significance between fentanyl and isoflurane groups. The bottom graph shows the distribution of isoflurane percentage for the anesthetic groups with the number of measurements (of approx. one-hour length each). Isoflurane percentages were recorded manually every 15 minutes, and an average across one recording (approx. 90 minutes of unchanged ICP baseline) was calculated.

autoregulatory dysfunction. When ICP oscillations were induced, a steady decrease in phase difference from the very first frequency of 0.009 Hz was found in the fentanyl group, which is a different behavior compared to ABP oscillations. For the isoflurane group, oscillations in ICP induced a phase difference between ABP and ICP which is close to 0 degrees (Fig 6, right), indicating that CA might be compromised.

## Discussion

Our results show that measurements of static, dynamic, and pseudo-dynamic autoregulation under anesthesia induced autoregulation impairment are in general agreement. We further determined that CPP is a better comparison value of CA than ABP or ICP alone, and that ABP and ICP oscillations result in different yet comparable frequency responses of pressure reactivity. These findings are further discussed below.

Typical measurements of CA rely on perturbing the autoregulatory system and measuring the response. Our perturbations were based on a height change in a saline reservoir connected to the lateral ventricle and inducing oscillations around the baseline (ΔICP), and manipulation

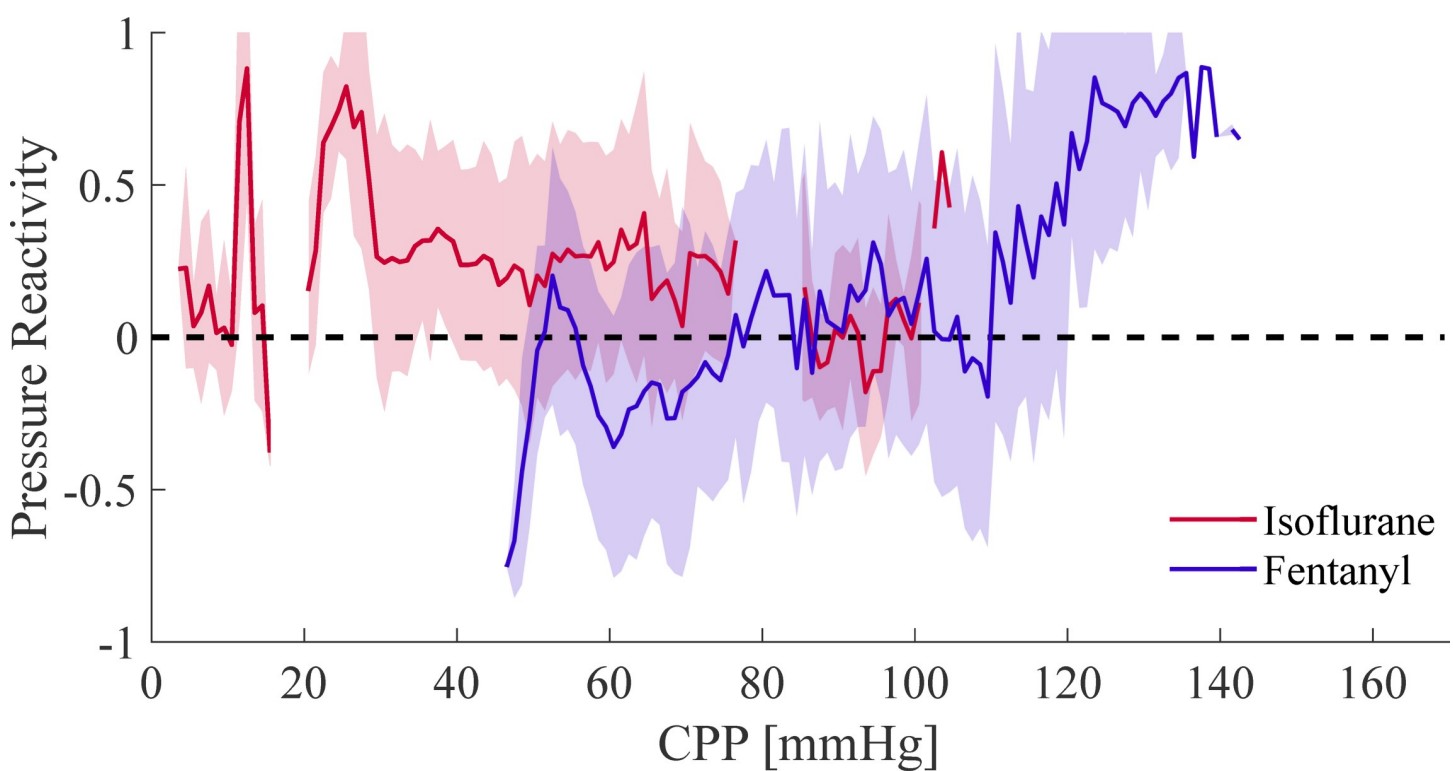

**Fig 5. Pressure reactivity (PRx) values plotted as averages according to underlying CPP values.** The red line indicates the averaged response of isoflurane anesthetized NHPs, while blue is the average of fentanyl. Shaded areas show the standard deviation. The dashed line marks a correlation of 0.

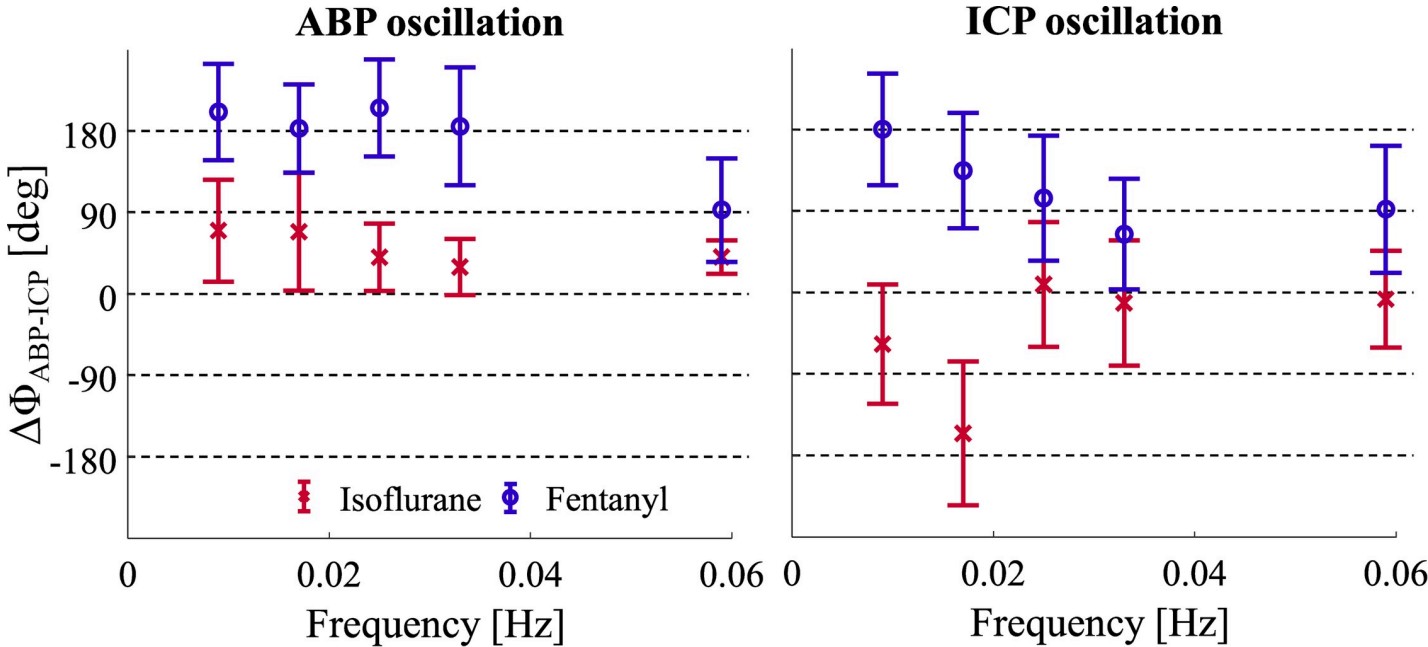

**Fig 6. The phase lag between ABP and ICP during oscillations in ABP (left) and ICP (right).** Red markers show isoflurane anesthesia group averages with error bars spanning the standard deviation. Blue markers show the fentanyl group.

of PEEP (ΔABP). We showed that oscillations in ΔCBF occur as a reaction to ICP and ABP oscillations (Fig 2). While a change in CBF alone is not an indicator of CA impairment, long term elevation of reduction in CBF is, which is captured by Lassen's curve. During high ICP, corresponding to low CPP, the LLA of the Lassen's curve (Fig 4) can be exceeded and CA becomes impaired. $CO_2$ is a known regulator for CBF and ABP. While we did see a positive correlation between ABP and $ETCO_2$, we did not observe $ETCO_2$ values were confined within a small range within normal limits (Fig 3). We therefore assume that $ETCO_2$ was not the leading cause of CBF changes, but rather a consequence of increased heart rate and ABP as an autoregulatory response to induced ICP changes. We evaluated CA impairment based on Static, Pseudo-Dynamic, and Dynamic CA, allowing us to capture and compare a range of CA measurement algorithms proposed in previous years. To control for a presumable ground truth of CA impairment in the NHPs, we have used two different protocols of anesthesia, namely higher percentage of isoflurane gas anesthetics (> 1.5%) for CA impairment and very low isoflurane (< 1%) in combination with predominantly fentanyl anesthesia, administered intravenously, for intact CA.

The ranges of CPP (Fig 4) between the two groups were significantly different (p = 0.005) and showed little overlap. This was likely an effect of the anesthetic itself as isoflurane has been shown to suppress ABP at higher dosage [29]. The wider range of CPP in the fentanyl group allowed for a full reconstruction of Lassen's curve. The reconstructed Lassen's curve followed the literature based canonical shape and clearly showed the plateau of intact autoregulation were ΔCBF was not changing. The isoflurane group with the reduced CPP showed most of the data beyond the LLA, showing a dependency of ΔCBF and CPP and autoregulatory impairment. This observation was confirmed by the PRx value (Fig 5), in which low correlation values indicate intact CA. Our results indicate that autoregulation is intact when using fentanyl, with PRx values below 0, while isoflurane impairs autoregulation with values of PRx above 0, often showing PRx > 0.3 in the CPP range of 50 mmHg to 80 mmHg on the plateau

of Lassen's curve. This further indicates that the isoflurane group had impaired autoregulation. Furthermore, the phase difference between ABP and ICP during pressure oscillations showed that dynamic autoregulation was impaired in isoflurane measurements for both ICP and ABP oscillations, as indicated by a smaller phase lag, and intact in fentanyl anesthesia, given the almost 180-degree phase difference. Given that all three CA approaches indicate the same trends, and that they are in accordance with the original publications and the anesthesia comparison in patients undergoing elective surgery given by Tiecks et al. [11], we are confident that CA manipulation was successful.

The results, however, differ from previous work significantly in that they were performed on healthy NHPs, not hospitalized patients. More importantly, the driving force for CPP changes in this study was a combination of changes in ICP and ABP. These two changes combined give weight to the hypothesis that it is not ABP that drives CA activity, but rather CPP. In this case, ICP plays an important role in the regulation of brain perfusion and has strong implications on treatment of patients at risk of elevated ICP, as seen in traumatic brain injury or hydrocephalus. This hypothesis is further strengthened by the similar reactions to oscillations at different frequencies in ICP and the frequency dependent response in case of intact CA during fentanyl anesthesia. Yet another indicating factor is the distribution of ICP and ABP values across the two anesthetic groups. While the CPP and ABP histograms showed a significant difference, ICP histograms for the same data set showed very similar distributions (as determined by Mann-Whitney U test in Fig 4). Furthermore, the overlap region of CPP values was smaller than the overlap in ABP. This shows that it is not the individual ABP or ICP value but their combination (CPP = mean ABP–ICP) that should be considered when measuring CA impairment.

Our observation of impaired autoregulation under isoflurane anesthesia in static and dynamic CA is shared with early findings from Strebel et al. [37]. They measured the impairment of dynamic CA through thigh cuff occlusion, finding the rate or regulation in transcranial Doppler measurements of the middle cerebral artery to be reduced. However, they report that static CA is only marginally changed, indicating that it is still intact, which differs from our observations. Another study by Summors et al. [38] demonstrated a severe loss in the rate of regulation, measured in the same way as Strebel et al., reporting a loss of dynamic CA under isoflurane. Our approach differed from these findings by firstly by measuring the microvasculature through DCS, rather than relying on cascading reactions to the major arteries. Secondly, we compared the influence of ABP and ICP independently under the conditions of different anesthetics. The two different pressure challenges have shown to differ in their response, while reaching similar conclusions (see Fig 6).

Some characteristics of this study should be pointed out to evaluate the importance of the results. First, the use of an animal model on healthy NHPs, in which similarity to human anatomy and physiology is high, might still show differences in auto-regulatory abilities in a direct comparison to healthy human subject. Without reliable non-invasive assessment of ICP, this limitation cannot be easily overcome. Second is the inter-subject variability, where variation in probe placement and rate of anesthetic should be considered. We used stereotaxic coordinates to place our measurement probe and cannula in an equivalent location across all NHPs. However, slight variations are possible, which can progress to the placement of the DCS probe to measure ΔCBF, which was placed relative to the pressure probe and needed to be fitted to the skull's shape. The rate of anesthetic administered for both fentanyl and isoflurane varied across NHPs, due to differences in weight and metabolism, and sometimes had to be adjusted during the experiment. Therefore, the degree of CA impairment between NHPs within a group might vary slightly. Lastly, the values of ABP and ICP rely on assumptions. ABP was measured in the carotid artery and assumed to be systemic in this work, but ABP inside the skull might vary.

Similarly, ICP was assumed to be global across the cranium. This assumption is not necessarily valid in clinical applications as pressure distributions can potentially vary with lesions, fractions, or tumors.

## Conclusions

The results presented here show that isoflurane strongly impairs CA, in both dynamic and static CA measurement approaches, while fentanyl anesthesia allows for adequate pressure and cerebrovascular flow regulation. The impairment applied to both ICP and ABP triggered perturbations of the vascular system, suggesting that it is the CPP value that sets CA in motion. This is further supported by observations that ICP distributions can be similar while ABP and especially CPP distributions between fentanyl- and isoflurane- anesthetized NHPs are significantly different. This implies that patient treatment should potentially be driven by CPP regulation in diseases such as hydrocephalus, stroke, and traumatic brain injury, instead of relying on ABP or ICP alone.

## Acknowledgments

The authors would like to thank ZOLL Medical Corporation (Chelmsford, MA, USA), for lending the programmable ventilator.

## Author Contributions

**Conceptualization:** Alexander Ruesch, Deepshikha Acharya, Matthew A. Smith, Jana M. Kainerstorfer.

**Data curation:** Alexander Ruesch, Deepshikha Acharya, Samantha Schmitt, Jason Yang, Jana M. Kainerstorfer.

**Formal analysis:** Alexander Ruesch, Samantha Schmitt, Jason Yang.

**Funding acquisition:** Matthew A. Smith, Jana M. Kainerstorfer.

**Investigation:** Alexander Ruesch, Matthew A. Smith, Jana M. Kainerstorfer.

**Methodology:** Alexander Ruesch, Deepshikha Acharya, Samantha Schmitt, Jason Yang, Matthew A. Smith, Jana M. Kainerstorfer.

**Project administration:** Samantha Schmitt, Matthew A. Smith, Jana M. Kainerstorfer.

**Resources:** Matthew A. Smith, Jana M. Kainerstorfer.

**Software:** Alexander Ruesch, Jason Yang.

**Supervision:** Matthew A. Smith, Jana M. Kainerstorfer.

**Validation:** Alexander Ruesch, Deepshikha Acharya, Samantha Schmitt, Jason Yang, Matthew A. Smith, Jana M. Kainerstorfer.

**Visualization:** Alexander Ruesch.

**Writing – original draft:** Alexander Ruesch.

**Writing – review & editing:** Alexander Ruesch, Deepshikha Acharya, Samantha Schmitt, Matthew A. Smith, Jana M. Kainerstorfer.

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
