## [Decision Letter · Decision Letter 0]

8 Oct 2020

PONE-D-20-28572

Comparison of static and dynamic cerebral autoregulation in a controlled animal model

PLOS ONE

Dear Dr. Alexander Ruesch,

Thank you for submitting your manuscript to PLOS ONE. After careful consideration, we feel that it has merit but does not fully meet PLOS ONE’s publication criteria as it currently stands. Therefore, we invite you to submit a revised version of the manuscript that addresses the points raised during the review process.

The subject of this study is still controversial and it should bring new insights into this issue. Since, however, the authors do not provide sufficient information, it is difficult properly to evaluate this study. As the reviewers pointed out, additional data and detailed explanations are required. Furthermore, the novelty of this study must clearly be emphasized.

We look forward to receiving your revised manuscript.

Kind regards,

Yoko Hoshi, M.D. Ph.D.

Academic Editor

PLOS ONE

Journal Requirements:

Reviewers' comments:

Reviewer's Responses to Questions

**Comments to the Author**

1. Is the manuscript technically sound, and do the data support the conclusions?

Reviewer #1: Yes

Reviewer #2: Yes

2. Has the statistical analysis been performed appropriately and rigorously? 

Reviewer #1: Yes

Reviewer #2: Yes

3. Have the authors made all data underlying the findings in their manuscript fully available?

Reviewer #1: Yes

Reviewer #2: Yes

4. Is the manuscript presented in an intelligible fashion and written in standard English?

Reviewer #1: Yes

Reviewer #2: Yes

5. Review Comments to the Author

Reviewer #1: There are several areas which should be clarified by the authors:

1. No data for arterial O2 and CO2 are provided. This is key as CO2 is a powerful regulator of CBF. Given the length of the preparations (16.2 h), it is probable that atelectasis to some degree occurred due to prolonged supine placement. Data for O2 and CO2 MUST be provided to validate the conclusions drawn.

2. The animals were paralyzed. Why? CO2 level will vary as depth of paralysis varies. How was such depth of paralysis monitored? Again, it is imperative to provide data for CO2 to properly interpret and confirm conclusions regarding effects on CA determined in this study. IACUC often forbids use of paralysis without clear strong proper justification.

3. This paper appears a bit improperly framed. It really is a study of anesthetics and their effects on CA. In particular, the authors appear quietly to advocate use of anesthetics to test CA intactness instead of other typical techniques (thigh cuff) for which they feel there is un necessary pain/risk. Perhaps, therefore, the title of the MS should be altered to reflect this agenda and the corresponding study hypothesis re-articulated to emphasize that this is a study of anesthetics and CA. If so, this paper really should be published in an anesthesia journal.

4. Going along with the above, the authors do not provide proper documentation that many others have studied the effects of anesthetics on CA. The authors should provide an enhanced discussion on this topic. If properly framed, then, what is really new here? Certainly, the experiments were conducted rigorously with newer techniques for CA determination (DCS), but truly the idea that anesthetics affect CA is not at all new. This paper appears to only incrementally add to the fund of knowledge in the literature.

5. What is the power for these studies? The n is quite small (7 and 5) for the 2 experimental groups. Only males were used with no justification. Therefore, these studies are not at all consistent with present emphasis on Rigor. In particular, this is important since LLA and CA may be dependent on gender.

Reviewer #2: The manuscript was described well except for the following points. Please rewrite these points properly.

1. The paragraph starting with; “Static Autoregulation” (P8L23-P9L8), is difficult to understand how to observe data. Please rephrase the whole paragraph with clear explanations for the following terms and phrases; “beta-value”, “Laser instabilities”, “intensity auto-correlation at zero-delay time”, “beta started fluctuating” and “alpha Db value”.

2. It is difficult to understand a new term of “Pseudo-Dynamic” autoregulation. The concept for this newly defined phrase must be explained.

Most legends for the Figures are insufficient to understand contents well because of poor definition or poor drawing of x-axis and y-axis.

1. Fig.1: Need explanation in the legend how to control and measure airway pressure for and how to this is transferred to MAP, and also how to control saline reservoir sinusoid way.

2. Fig.2: Need explanation for zoom-up on the right, how to obtain sinusoidal curves of the ICP. It is unclear why the ABP follows the ICP because the ABP must be independent of ICP. Describe how the CPP was measured.

3. Fig.3: Need explanation for histograms of CPP, ABP and ICP. In the bottom graph, need explanation how to calculate the measurement for each isoflurane percentage.

4. Fig.4: Do not hide the representative line for isoflurane under the fentanyl SD curves.

In addition, several terminologies were not defined properly. Need clear definition for the following terms in main text.

1. PRx (pressure reactivity index) (P4L11)

2. PEEP (positive end-expiratory pressure) oscillation (P5L5)

3. alphaDb (diffuse coefficient) (P6L6)

6. PLOS authors have the option to publish the peer review history of their article (what does this mean?). If published, this will include your full peer review and any attached files.

Reviewer #1: No

Reviewer #2: No

---

## [Author Response · Author response to Decision Letter 0]

22 Nov 2020

The response to the reviewers has been uploaded as a file, including a color code to identify changes and additions to the manuscript more easily. 

Please see the attached file.

Thank you.

---

## [Decision Letter · Decision Letter 1]

18 Dec 2020

PONE-D-20-28572R1

Comparison of static and dynamic cerebral autoregulation under anesthesia influence in a controlled animal model

PLOS ONE

Dear Dr. Alexander Ruesch,

Thank you for submitting your manuscript to PLOS ONE. After careful consideration, we feel that it has merit but does not fully meet PLOS ONE’s publication criteria as it currently stands. Therefore, we invite you to submit a revised version of the manuscript that addresses the points raised during the review process.

The editor would like to ask the authors to revise  Fig. 4 according to the reviewer's comment.

We look forward to receiving your revised manuscript.

Kind regards,

Yoko Hoshi, M.D. Ph.D.

Academic Editor

PLOS ONE

Reviewers' comments:

Reviewer's Responses to Questions

**Comments to the Author**

1. If the authors have adequately addressed your comments raised in a previous round of review and you feel that this manuscript is now acceptable for publication, you may indicate that here to bypass the “Comments to the Author” section, enter your conflict of interest statement in the “Confidential to Editor” section, and submit your "Accept" recommendation.

Reviewer #1: All comments have been addressed

Reviewer #2: (No Response)

2. Is the manuscript technically sound, and do the data support the conclusions?

Reviewer #1: Yes

Reviewer #2: Partly

3. Has the statistical analysis been performed appropriately and rigorously? 

Reviewer #1: Yes

Reviewer #2: I Don't Know

4. Have the authors made all data underlying the findings in their manuscript fully available?

Reviewer #1: Yes

Reviewer #2: Yes

5. Is the manuscript presented in an intelligible fashion and written in standard English?

Reviewer #1: Yes

Reviewer #2: Yes

6. Review Comments to the Author

Reviewer #1: The authors have provided appropriate address of prior comments and have made suitable revisions in the new MS.

Reviewer #2: The MS revised almost properly except for the comment on Fig.4. This reviewer requested to clearly plot average lines for two anesthesia, Isoflurane and Fentanyl. However, Fig.4 has not been revised in the revised MS. Please revise Fig.4 as this reviewer requested, because average lines are important to evaluate the static autoregulation.

Otherwise, MS is properly revised.

7. PLOS authors have the option to publish the peer review history of their article (what does this mean?). If published, this will include your full peer review and any attached files.

Reviewer #1: No

Reviewer #2: No

---

## [Author Response · Author response to Decision Letter 1]

22 Dec 2020

Please see now attached document for response to the reviewers.

---

## [Editor Report · Decision Letter 2]

26 Dec 2020

Comparison of static and dynamic cerebral autoregulation under anesthesia influence in a controlled animal model

PONE-D-20-28572R2

Dear Dr.Alexander Ruesch ,

We’re pleased to inform you that your manuscript has been judged scientifically suitable for publication and will be formally accepted for publication once it meets all outstanding technical requirements.

Kind regards,

Yoko Hoshi, M.D. Ph.D.

Academic Editor

PLOS ONE
---

## [Editor Report · Acceptance letter]

2 Jan 2021

PONE-D-20-28572R2 

Comparison of static and dynamic cerebral autoregulation under anesthesia influence in a controlled animal model 

Dear Dr. Ruesch:

I'm pleased to inform you that your manuscript has been deemed suitable for publication in PLOS ONE. Congratulations! Your manuscript is now with our production department. 

Kind regards, 

on behalf of

Dr. Yoko Hoshi 

Academic Editor

PLOS ONE